# Critical ionic transport across an oxygen-vacancy ordering transition

Ji Soo Lim[1,2], Ho-Hyun Nahm[1], Marco Campanini [3], Jounghee Lee[1], Yong-Jin Kim[1,2], Heung-Sik Park[1,2], Jeonghun Suh[1,2], Jun Jung [1], Yongsoo Yang [1], Tae Yeong Koo[4], Marta D. Rossell [3] ✉, Yong-Hyun Kim [1,5] ✉ & Chan-Ho Yang [1,2,6] ✉

Phase transition points can be used to critically reduce the ionic migration activation energy, which is important for realizing high-performance electrolytes at low temperatures. Here, we demonstrate a route toward low-temperature thermionic conduction in solids, by exploiting the critically lowered activation energy associated with oxygen transport in Ca-substituted bismuth ferrite ($Bi_{1-x}Ca_xFeO_{3-\delta}$) films. Our demonstration relies on the finding that a compositional phase transition occurs by varying Ca doping ratio across $x_{Ca} \simeq 0.45$ between two structural phases with oxygen-vacancy channel ordering along <100> or <110> crystal axis, respectively. Regardless of the atomic-scale irregularity in defect distribution at the doping ratio, the activation energy is largely suppressed to 0.43 eV, compared with ~0.9 eV measured in otherwise rigid phases. From first-principles calculations, we propose that the effective short-range attraction between two positively charged oxygen vacancies sharing lattice deformation not only forms the defect orders but also suppresses the activation energy through concerted hopping.

Lattice defects such as interstitials and vacancies are omnipresent in crystalline solids and exploiting their mobility is a key factor for future applications such as batteries, fuel cells, resistive switching memories, electrochromic cells and neuromorphic devices[1–10]. Defect migration is typically depicted as a single particle hopping from an initial site to a final site through a rigid lattice structure, so, the general design principle for attaining high ionic conduction has been to find more spacious paths and interfaces[11,12]. Much effort has been put into optimizing materials to reduce the migration activation energy ($E_A$) required for low-temperature ($T$) operation[1–3,7,11,12]. The typical values of $E_A$ for good oxygen conductors, such as yttria-stabilized zirconia used as oxygen electrolytes in solid oxide fuel cells, are approximately 0.7 eV (refs. 1, 2). Although lower $E_A$ values in the range of 0.4–0.6 eV have also been experimentally measured, such as in barium-containing cobalt

iron oxides[8,13], BIMEVOX series[1], and so on, practical applications demand material durability and stable ion-dominant conduction. The quest of optimizing low $E_A$ as well as discovery of new materials are continuously required to pursue superionic conduction at low $T$s.

Here, we propose a way to significantly lower $E_A$ by controlling chemical doping to find the compositional phase transition point that separates two different structural phases of oxygen-vacancy ($V_O$) order. Transition metal oxides provide promising platforms for implementing the idea because high-density $V_O$s can be spontaneously produced owing to the high solubility of chemical substitution and the inherent valence nature of transition metals[14,15] and various $V_O$ orderings exist, as in aliovalent doped ferrites such as lanthanum strontium ferrites (LaSr)FeO$_{3-\delta}$ (refs. 3, 16) and the Ca-doped BiFeO$_3$ ($Bi_{1-x}Ca_x$FeO$_{3-\delta}$; BCFO)[17–23] which is examined in this study.

[1]Department of Physics, Korea Advanced Institute of Science and Technology (KAIST), Yuseong-gu, Daejeon 34141, Republic of Korea. [2]Center for Lattice Defectronics, KAIST, Yuseong-gu, Daejeon 3414, Republic of Korea. [3]Electron Microscopy Center, Empa, Überlandstrasse 129, Dübendorf 8600, Switzerland. [4]Pohang Accelerator Laboratory, POSTECH, Pohang, Gyeongbuk 37673, Republic of Korea. [5]Graduate School of Nanoscience and Technology, KAIST, Yuseong-gu, Daejeon 34141, Republic of Korea. [6]KAIST Institute for the NanoCentury, KAIST, Yuseong-gu, Daejeon 34141, Republic of Korea. ✉e-mail: marta.rossell@empa.ch; yong.hyun.kim@kaist.ac.kr; chyang@kaist.ac.kr

BCFO is known as a compensated semiconductor due to the balanced coexistence of $V_O$ donors and Ca acceptors ($\delta = x_{Ca}/2$) while maintaining the valence state of $Fe^{3+}$ (ref. 17). Even if efforts were made to further oxidize the compounds by thermal annealing at a high oxygen pressure of 125 bar at a high temperature of 800 °C, the variation of $V_O$ content turned out to be quite small[22]. Naturally produced $V_O$s are confined into oxygen-deficient layers similar to those shown in the brownmillerite or Grenier phases (however, one of the four $V_O$s is filled by oxygen ion and the details will be addressed in the theoretical part), establishing defect superlattice structures in a self-assembled way[19]. The increase of $x_{Ca}$ produces proportionally more planar defect layers, thereby reducing the average distance ($d$ in monolayers) between adjacent oxygen-deficient layers ($d \simeq 1.5/x_{Ca}$). Regardless of the $x_{Ca}$ ratio, the density of $V_O$s within the planar defect layer remains constant, but the change in the Bi/Ca ratio controls the crystallographic $c/a$ ratio. On these grounds, the BCFO system is an ideal platform to explore the phase transition of planar defect ordering within a two-dimensional oxygen-deficient layer by controlling the $c/a$ ratio.

## Results

### Oxygen migration and $E_A$ lowering

The atomic-scale hopping of thermally excited $V_O$ ions will statistically create a net drift flow in inhomogeneous electrical and strain fields, and/or induce a net diffusional flow due to the non-uniform distribution of defect concentration. Since the structural, electronic, and optical properties of solids are greatly affected by the concentration of defects[17,23], observing color propagation on a ten-micron length scale offers the phase evolution and the microscopic kinetic quantities of defects[18]. In a fabricated channel (400-µm-long and 50-µm-width) of a high-quality BCFO thin film (Supplementary Fig. 1), positively ionized $V_O$s moved in electric fields and piled up near the ground electrode (Fig. 1a). When the mobile donors in a local area are electrically removed, the electrically-formed region optically becomes darker in color, being electronically hole-doped. The electrocoloration is an established approach for examining defect kinetics in compensated semiconductors, and the technique was applied to acceptor-doped $SrTiO_3$ and the development of $n$-type, ionic, and $p$-type regions was carefully analyzed[24,25].

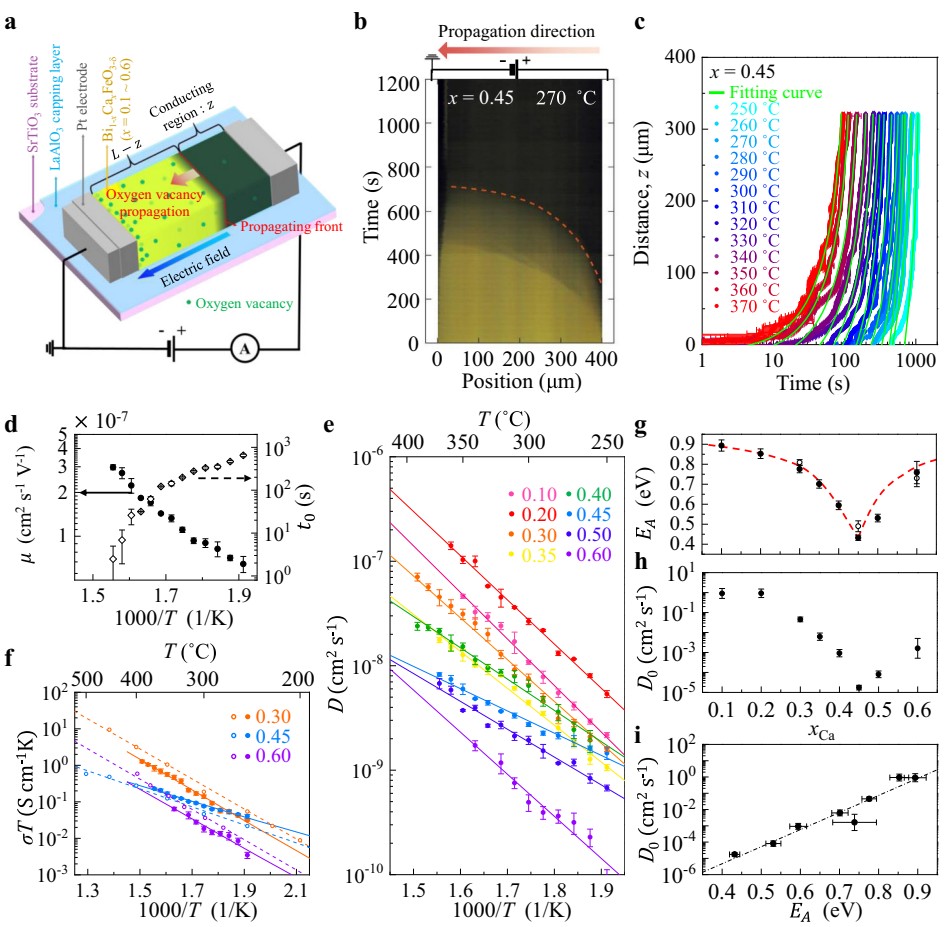

**Fig. 1 | Real-time observation of $V_O$ migration in BCFO films. a** Schematic of a BCFO channel protected by a $LaAlO_3$ capping layer with a pair of 400-µm gapped coplanar electrodes. Optical microscopic video and real-time current were simultaneously monitored while a constant voltage was applied. $V_O$s migrated toward the ground electrode, thereby extending a conducting dark-colored formed phase. **b** Time evolution of the optical contrast along the centerline of the channel during the electroforming process at $x_{Ca}= 0.45$. The orange dashed line represents the boundary between the intermediate dark-yellow phase and the completely-formed dark-colored phase. **c** Trajectories of the boundary position as a function of elapsed time at selected $T$s. Horizontal and vertical error bars represent the full-width-at-half-maximum (FWHM) of differential color change Gaussian profiles along the position and time coordinates. The green fitting curves indicate that the simplified model (described in the main text) matches the experimental data well. **d** The fitting variables of ionic mobility and time offset were evaluated from the fitting in **c**. **e** Arrhenius plots of $V_O$ diffusivity in BCFO films of different $x_{Ca}$s (0.1 - 0.6). Solid lines indicate the linear relationship between logarithmic diffusivities and inverse $T$s. **f** Arrhenius plots of ionic conductivity times $T$. Filled dots were obtained using the optical visualization. Empty dots were obtained using AC impedance spectroscopy of $N_2$ gas annealed BCFO films. **g** The $E_A$ of $V_O$ diffusivity with respect to $x_{Ca}$. Both filled dots (from the optical visualization) and empty dots (from the AC impedance spectroscopy) are well matched on the red guide line. **h** Prefactor of diffusivity ($D_0$) as a function of $x_{Ca}$. **i** Correlation between logarithmic $D_0$ and linear $E_A$. Error bars in **d**–**i** represent the standard errors of fittings or the values calculated from them based on error propagation for the derived parameters.

We isolated the optical contrast along the central horizontal line of a $Bi_{0.55}Ca_{0.45}FeO_{3-\delta}$ (BCFO at $x_{Ca} = 0.45$) channel to construct a color-evolution diagram (Fig. 1b; a full set of diagrams at different $T$s are in the Supplementary Fig. 2). To quantitatively analyze the propagation of the color boundary between the intermediate and final phases, we digitalized the trajectory of the boundary as a function of elapsed time, as shown in Fig. 1c (see also the Supplementary Note 1). The trajectory curves can be described by an equation (see Methods section for video imaging analysis), $z(t) = L - \sqrt{L^2 - 2\mu V(t - t_0)}$, ($z(t)$: the length of the conducting dark phase, $L$: the channel length, $t$: an elapsed time), on the assumption that an electric bias $V$ was applied entirely across the more insulating region with length $L - z$. The other two terms, $\mu$ and $t_0$, are ionic mobility and time offset, determined from theoretical fitting curves (green line in Fig. 1c). The determined $\mu$ and $t_0$ are plotted as a function of $1/T$ (Fig. 1d).

Ionic conduction diffusivity ($D$) can be also calculated from $\mu$, according to the Nernst–Einstein relation $D = \mu k_B T / q$, where $k_B$ is Boltzmann constant and $q$ stands for ionic charge $2e$. A more rigorous form of the chemical diffusion coefficient ($\tilde{D}$) has been developed through consideration of the defect correlation existent in such high-density defect systems, as discussed by G. E. Murch[26]. The interaction effect can be considered by multiplying the classical form by a thermodynamic enhancement factor ($W$). In case that electronic conductivity predominates the ionic one and the electroneutrality condition is satisfied, the $W$ can be approximated to $1 + \frac{q}{e}\frac{n_i}{n_e}$, explicitly depending on the ratio of ionic concentration to electronic one (ref. [27]). Therefore, the classical Nernst–Einstein relation gives an underestimated value of $\tilde{D}$ by a factor of $W$ presumed to be ~2 or ~3 compared to the value measured by electrochemical impedance spectroscopy (Supplementary Note 2).

An investigation of $D$ with varying $T$ and $x_{Ca}$ (Fig. 1e; each data point can be obtained by independent video recording and performing the quantification/fitting procedure) shows a linear relationship of $D$ versus $1/T$, revealing the barrier $E_A$ of oxygen migration. The ionic conductivity, $\sigma = nq\mu$, estimated using the defect density of total $V_O$s, $n$, remarkably matches the values obtained from standard AC impedance spectroscopy for nitrogen-annealed samples, in which oxygen ion conduction is dominant over electronic conduction (Fig. 1f, Supplementary Fig. 3 and Supplementary Note 3). Besides, the tracer diffusion experiment using an oxygen isotope $^{18}O$ gas on a BCFO film at $x_{Ca} = 0.45$ directly exhibits the oxygen ion lateral conduction with a similar value of diffusivity (Supplementary Fig. 4 and Supplementary Note 4). Interestingly, as $x_{Ca}$ increased, the $E_A$ value became lower. A minimum value of 0.43 eV could be achieved at $x_{Ca} = 0.45$, which is a notably low value compared to conventional oxide ionic conductors[1,2] and comparable to the values of Ba-contained cobaltates[8,13]. When $x_{Ca}$ exceeded 0.45, the $E_A$ increased, exhibiting a symmetric shape around $x_{Ca} = 0.45$ (Fig. 1g).

The prefactor of diffusivity ($D_0$) showed a behavior similar to the $E_A$ (Fig. 1h). As a result, the logarithmic $D_0$ is linearly proportional to $E_A$ (Fig. 1i), which is known as the Meyer–Neldel rule. The diffusion coefficient is expressed as $D = \frac{1}{4}a^2 Z \upsilon^0 \exp\left(\frac{\Delta S_{mig}}{k_B}\right)\exp\left(\frac{-\Delta H_{mig}}{k_B T}\right)$, ($a$: the jumping distance, $Z$: the number of neighboring sites, $\upsilon^0$: the attempt frequency). The changes in migration entropy $\Delta S_{mig}$ and enthalpy $\Delta H_{mig}$ come from the difference in Gibbs free energy between the two configurations, where oxygen ions are located at equilibrium sites or unstable saddle points[28]. The $E_A$ measured at a constant $T$ and pressure corresponds to $\Delta H_{mig}$. $\Delta S_{mig}$ is usually in the range of $1–2k_B$ (refs. [29], [30]). The large suppression of $D_0$ at $x_{Ca} = 0.45$ as much as $10^5$ suggests an abnormally small value of $\Delta S_{mig} \sim -10k_B$ and/or a significantly low vibration frequency $\upsilon^0$.

## $V_O$ ordering transition

Materials are made up of many interacting ions. In principle, ionic motion occurs in a multidimensional energy landscape where the migration path to the barrier involves configuration changes. Figure 2a describes the essence of our picture regarding the phase transition point between two structural phases with different $V_O$ orderings along either <100> or <110> directions. As depicted in the free energy landscapes, atomic arrangement configurations have similar energies regardless of the somewhat irregular defect orders and short-range mixtures at the compositional phase transition point. This is in contrast to the lower and higher doping regimes, where well-defined channel configurations exist as ground state configurations. More two-dimension (2D)-like ionic conduction is expected in the sample at the phase transition point, due to greater flattening of the energy landscape, which is distinct from ion flow along rigid one-dimensional (1D) channels.

Confirmation of the presence of ordered $V_O$ channels in the BCFO films ($x_{Ca} = 0.3$, 0.45, and 0.6) was obtained by electron diffraction (ED, Supplementary Note 5) and scanning transmission electron microscopy (STEM). The quantitative analysis of the lattice distortions in the atomic-scale annular-bright-field (ABF) STEM images (Supplementary Figs. 5 and 6) demonstrates the presence of oxygen-deficient planar layers, which appear periodically every ~6 ($x_{Ca} = 0.3$), 4 ($x_{Ca} = 0.45$), as well as 3 ($x_{Ca} = 0.6$) perovskite blocks along the [001] growing direction.

The variations in the intensity of the oxygen columns in the ABF images along the [010] direction are used to reveal the position of the $V_O$s (Fig. 2b). Intensity line profiles extracted along the in-plane direction in the oxygen-deficient layers illustrate the Fe/O–O configuration. Specifically, in the BCFO film at $x_{Ca} = 0.3$, every second oxygen column has mostly vanished, proving that $V_O$ channels running along the [010] direction alternate with fully occupied oxygen columns along the in-plane [100] direction. Unlike the BCFO at $x_{Ca} = 0.3$, the films at $x_{Ca} = 0.45$ or 0.6 exhibited significantly smaller differences in intensity between oxygen columns, suggesting that the $V_O$ channels are either incoherently established along the [010] direction or aligned along another direction (a quantitative analysis is provided in Supplementary Fig. 7a, b).

Along the $[\bar{1}10]$ zone axis, direct evidence of the occurrence of $V_O$ channels cannot be attained from the intensity variations of the oxygen columns. For the BCFO films with higher Ca content ($x_{Ca} > 0.45$), the oxygen-deficient layers consist of $V_O$ channels aligned parallel to the $[\bar{1}10]$ direction alternating with highly distorted chains of vertex-linked tetrahedra. In these tetrahedral chains, the tetrahedra undergo a cooperative rotation around [001] as in the brownmillerite structure[31,32]. As a result, the oxygen atoms in the oxygen-deficient layers are ordered in a zigzag fashion along the $[\bar{1}10]$ direction and partially overlap with the Fe atomic columns. The position of the $V_O$ channels can be instead inferred from the Fe–Fe distances, as the Fe cations nearest to the $V_O$ move slightly back into the remaining oxygen atoms to achieve tetrahedral coordination. The ABF-STEM images along the $[\bar{1}10]$ zone axis (Fig. 2c) and the resultant line profiles of the BCFO at $x_{Ca} = 0.6$ reveal that two distinct Fe–Fe distances alternate along the in-plane [110] direction, while less pronounced distance differences are observed for the BCFO films at $x_{Ca} = 0.3$ and 0.45, suggesting the $V_O$ channels are not well established along the $[\bar{1}10]$ direction. Supplementary Fig. 7c provides a quantitative analysis of the Fe–Fe distances measured from the HAADF-STEM images.

In a complementary way, we carried out synchrotron-based X-ray diffraction for various reciprocal positions (Supplementary Fig. 8). As seen in the STEM results, the two-unit-cell periodicities of the $V_O$ channels along <100> or <110> directions can be identified by observation of <1/2 0 L> or <1/2 1/2 L> peaks in the HK reciprocal space maps (Supplementary Fig. 8a). Moreover, three L scans through the primary reflection or the in-plane half-ordered reflections offer versatile information of the out-of-plane stacking sequences of oxygen-deficient layers as well as the intra-layer structures (Supplementary Fig. 8b). To quantitatively explain the diffraction features, we perform

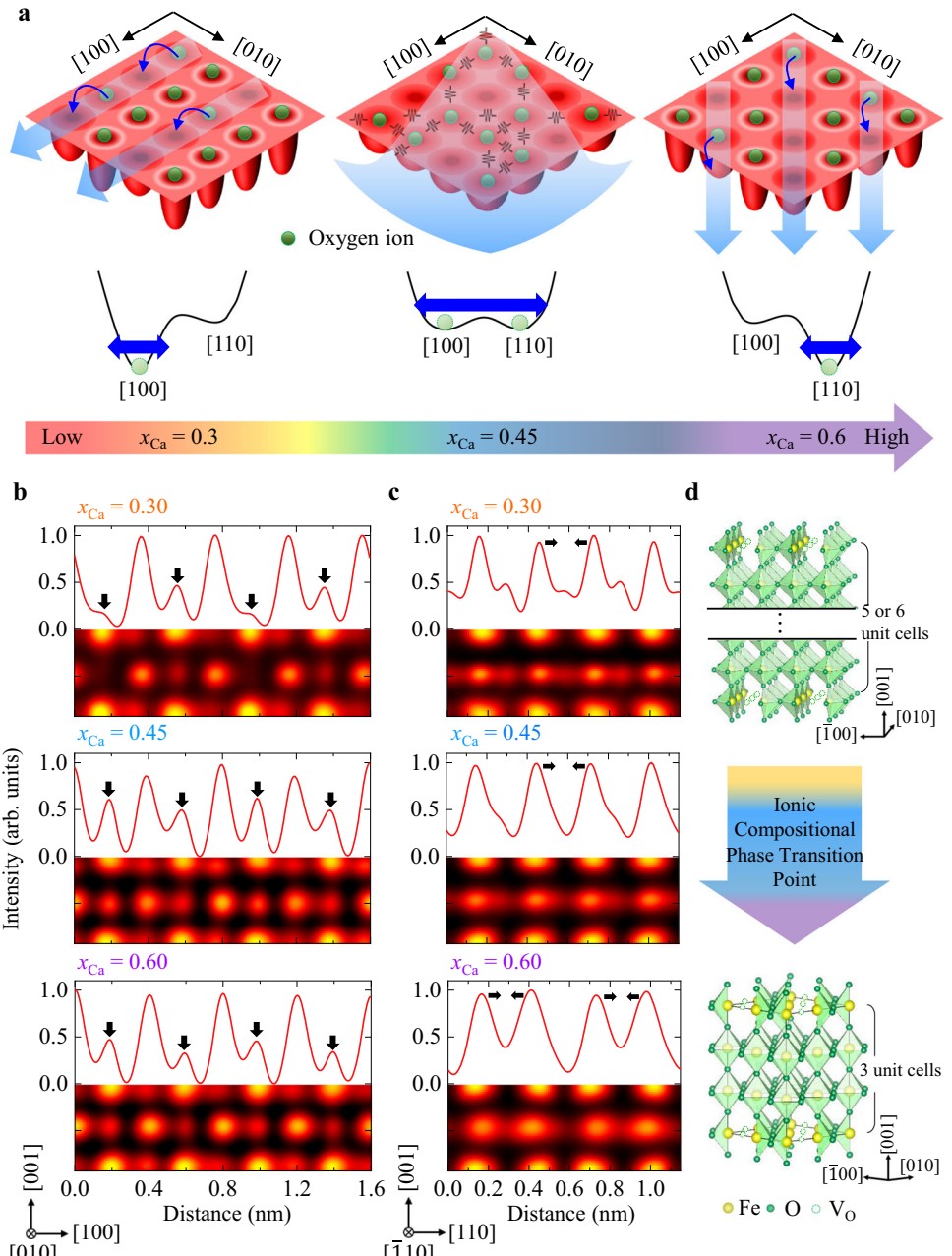

**Fig. 2 | Observation of $V_O$ channels and the compositional phase transition point in BCFO films. a** Schematics of three types of oxygen ion migration in 2D potentials. Oxygen ions avoid residing on the 1D atomic-scale channels because the potential therein is not relatively deep. Ionized $V_O$s are mainly present in the channel regions, which offer more positive potential in a self-consistent way. Free energy landscapes over the ionic configuration indicate competition between two types of channels along [100] or [110]. Configuration fluctuations can be significantly large at the phase transition point, leading to enhancement of generalized susceptibilities such as elastic compliance and ionic conductivity. **b**, **c** Representative ABF-STEM images along [010] ([$\bar{1}$10]) in the $V_O$ layers present in

BCFO ($x_{Ca} = 0.3$, 0.45 and 0.6), and corresponding intensity line profiles. The intensity line profiles averaged over 15 pixels were extracted from the ABF-STEM images along the [100] ([110]) direction, i.e., across the Fe/O−O atomic columns. The black arrows in **b** indicate that the intensity of the oxygen atomic columns is not uniform, instead $V_O$ ordering is observed. The black arrows in **c** show that the Fe–Fe distances are not uniform; instead two clearly different Fe–Fe distances alternate along the [110] direction. The Fe–Fe distances are $0.24 \pm 0.02$ and $0.33 \pm 0.02$ nm for the $x_{Ca} = 0.6$ phase, and $0.27 \pm 0.01$ and $0.30 \pm 0.01$ nm for the $x_{Ca} = 0.3$ and 0.45 phases. **d** Schematics of the atomic configuration of the BCFO films with $V_O$ channels.

structure factor calculations for the structural models depicted in Supplementary Fig. 8c (see the detailed structural parameters and domain population in the Supplementary Note 7). A spatial coherence between multiple $V_O$ domains arising from different stacking orders of the oxygen-deficient layers is essential for explaining the intensity modulation in the L scans. The nice matching between the experiment

and calculation confirms the competition of $V_O$ channel orders between <100> and <110>. The crystal structure with an oxygen order along <110> at $x_{Ca} = 0.6$ corresponds to the Grenier phase that has similar $V_O$ channels with the brownmillerite but the periodicity along $c$-axis is 3 (ref. 33). Furthermore, the somewhat complex $V_O$ structure at the compositional phase transition point seems to be a mixture

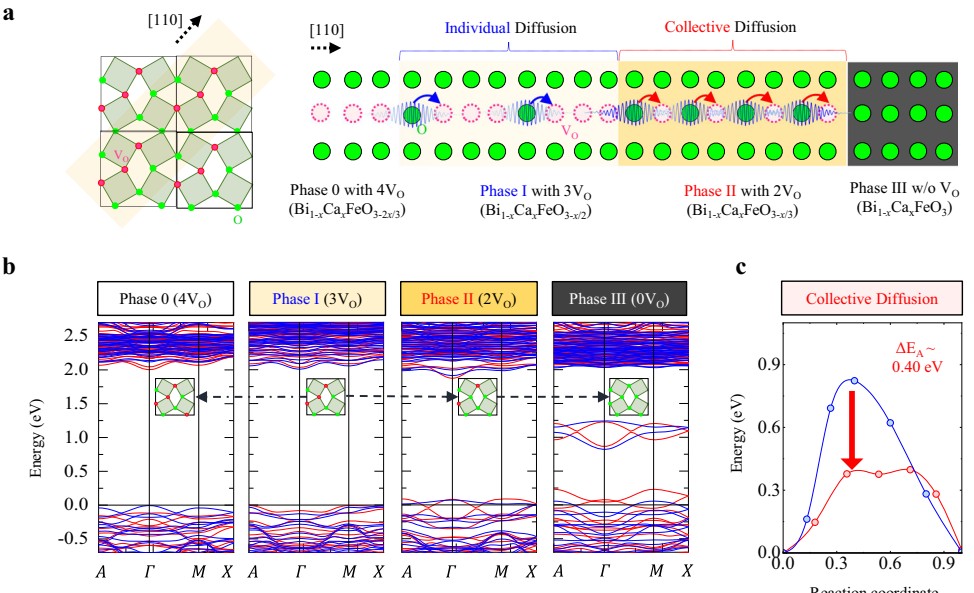

**Fig. 3 | DFT calculation results for collective ion diffusion. a** Schematic diagram of individual or collective diffusions of vibrating oxygen ions in the [110] channel of the phase I (as-grown BCFO with $3V_O$) and phase II (intermediate phase with $2V_O$). **b** Electronic structures of four possible phases, phase 0 with $4V_O$, as-grown phase I with $3V_O$, phase II with $2V_O$, and phase III without $V_O$, in a representative BCFO ($x_{Ca} = 0.25$) with fixing $n_{period} = 6$. Red and blue lines are spin-up and spin-down bands, respectively. Here, the reciprocal positions of the supercell are denoted by $A = (0.5, 0.5, 0.5)$, $\Gamma = (0.0, 0.0, 0.0)$, $M = (0.5, 0.5, 0.0)$, and $X = (0.5, 0.0, 0.0)$. **c** $E_A$ of $V_O$ migration for phases I and II. Both comparative calculations use periodic boundary conditions to mimic a sufficiently long coherence length of ions positional fluctuations (longer than the relaxation length). Only the phase II shows a significant reduction in $E_A$, indicating the high defect density in the intermediate phase, which appears in the nonequilibrium situation after electroforming, is a necessary condition for the correlative migration.

consisting of a zigzagging pattern as a result of the combination of [100] and [110] channels.

Based on these results, we conclude the $V_O$ channels are transmuted as $x_{Ca}$ increases, from [100] and [010] for $x_{Ca} = 0.3$ to [$\bar{1}$10] and [110] for $x_{Ca} = 0.6$ (Fig. 2d). This can be theoretically understood in the context of competing symmetric <100> and antisymmetric <110> tetrahedral arrangements. A smaller $c$-axis lattice constant with increasing $x_{Ca}$ causes a larger apical oxygen shift in the tetrahedron, as depicted in Supplementary Fig. 9f. In the symmetric <100> configuration, the two apical oxygens of adjacent tetrahedra are closer to each other than in the <110> case, resulting in a larger energy penalty. More quantitative energy comparisons between the two competing $V_O$-channel phases were studied at selected $x_{Ca}$ and in-plane lattice parameters (Supplementary Fig. 14).

**Possible collective transport mechanism**

The color change of the electroformed BCFO structures can be understood as sequential filling of three $V_O^{2+}$'s without changing the Ca concentration. Depending on the number of $V_O^{2+}$ in the oxygen-deficient plane, the BCFO transforms from a wide-gap transparent semiconductor (phase I: yellowish) to a metal with a smaller optical gap (phase II: dark-yellowish) and finally with in-gap hole states (phase III: black by hole polarons), as shown in Fig. 3 (phase 0 is just for comparison) (ref. 23). The observed propagation of the two color boundaries corresponds to the I/II and II/III phase boundaries, respectively.

Since the volume (and $c$-axis lattice parameter) of the deficient layer is significantly larger (more than 10%, see Supplementary Table 1) than the other layers, penetration of $V_O$ into other densely-packed layers rarely occurs. A similar theoretical consideration of ionic migration along the oxygen-vacancy channel has been explored in SrFeO$_{2.625}$, giving a significantly low calculated value (0.49 eV) of $E_A$ (ref. 34). In the SrFeO$_{2.625}$ structure, one of the four $V_O$ sites in the brownmillerite unit cell is occupied by oxygen ion. In that case, $V_O$ prefers to migrate in the in-plane channel direction to minimize the

repulsive $V_O$–$V_O$ interaction and to maintain stable tetrahedral layers within $V_O$ channels. So, we focused on the intra-layer hopping within the oxygen-deficient layer. We performed first-principles density functional theory (DFT) calculations for transient states along the migration paths to scrutinize whether the $E_A$ can be cooperatively lowered by simultaneously moving oxygen ions. We used periodic boundary conditions to mimic coherent transient states and hopping of multiple $V_O$s. In reality, thermal fluctuations create a decoherence that normally leads to random walking of individual ions. However, our hypothesis is that if the distance is small relative to some length scale, the positional deviations of the two ions at each equilibrium position are correlated and the correlation length can be exceptionally large at the critical point.

The calculated $E_A$ of the as-grown phase I (blue color of Fig. 3c) is ~0.82 eV, similar to the observed values in BCFOs out of the compositional phase transition point. However, the calculated $E_A$ of phase II (red color of Fig. 3c) is ~0.40 eV, which is much smaller than that of phase I. These indicate that the oxygen movement in phase II effectively takes energy gain through concerted hopping because the spacing between neighboring defects is less than the lattice relaxation length (Supplementary Figs. 9, 10 and Supplementary Note 6). Conclusively, the large suppression of $E_A$ at the compositional phase transition point of BCFO can be understood based on the cooperation of the inherent thermodynamic correlation length and the sufficiently close (phase II-like) interdistance of $V_O$ defects.

## Discussion

The attempt frequency $v^0$ might be proportionally related to the square root of the elastic modulus, of which the compositional dependence clearly indicates the phase transition point (Fig. 4). However, the $10^5$ times smaller $D_O$ at $x_{Ca} = 0.45$ cannot be explained by elastic softening alone. In the case of $V_O$ correlated diffusion, multiple ions are intimately linked leading to a significant reduction in the configuration dimension. A similar mechanism for lowering $E_A$ through multi-vacancy hopping has been proposed regarding nondilute Li

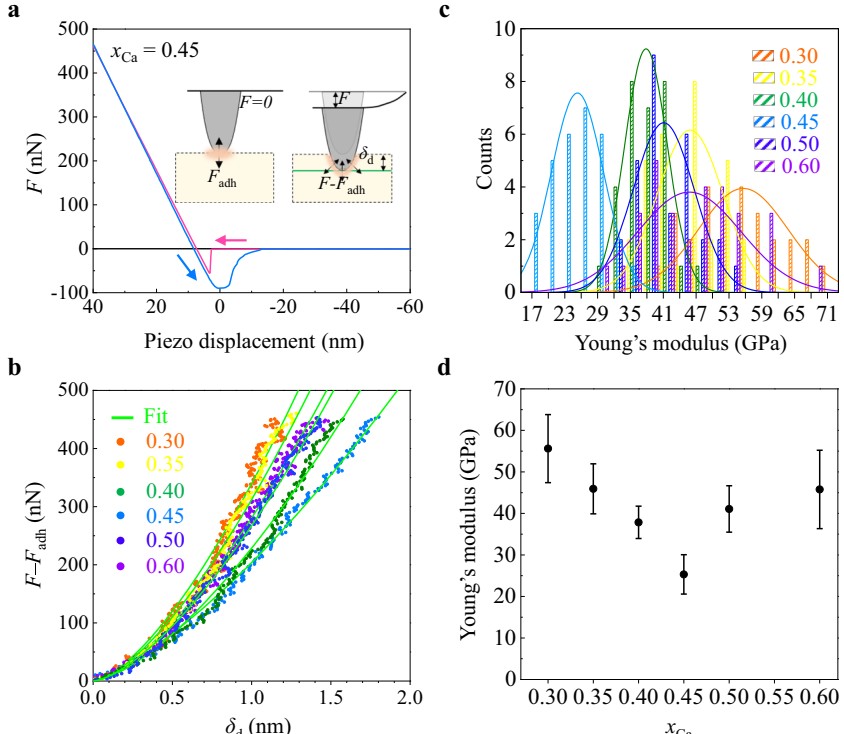

**Fig. 4 | Structural softness in the compositional phase transition point in BCFO films. a** Force versus distance curve of BCFO at $x_{Ca} = 0.45$, obtained by characterizing the elastic property using an AFM technique. (Inset) The measured force ($F$) was evaluated by cantilever bending against surface depth deformation ($\delta_d$). Zero cantilever deflection, i.e., $F = 0$, corresponds to the situation where the attractive adhesive force ($F_{adh} \leq 0$) between the sample and tip is fully compensated by the tip pressing force. So, the real tip-driven force on the sample surface is $F-F_{adh}$. Magenta and cyan indicate approach and retreat modes. **b** Tip-driven force ($F-F_{adh}$) versus $\delta_d$ curves for selected $x_{Ca}$. The DMT model was used to interpret these experimental data, creating green fit lines. **c** Statistics of the values of Young's modulus obtained by repeating the force-distance measurement at many different places on the surface. The lines represent Gaussian fittings of the observed distributions. **d** The measured Young's modulus versus $x_{Ca}$. The error bar is defined as the standard deviation of distributions.

diffusion[7,35]. Phase transition points have exhibited unexpected critical phenomena and the discussed concerted migration can be a functionality emerging at a phase transition point of the planar defect orders. More microscopic details beyond the phenomenological description need be clarified. In this context, we need to note the theoretical work of T. Das et al. reporting the progressively larger polaron volume with increasing Sr resulted in increasing oxygen-vacancy interactions in $La_{1-x}Sr_xFeO_{3-\delta}$ and a hypothetical rhombohedral phase at $x = 0.5$ has a higher oxygen $D$ than the other compositions[3].

In summary, we demonstrated oxygen transport with a substantially low diffusion barrier 0.43 eV near a compositional phase transition point in Ca-substituted bismuth ferrite thin films. The two competitive $V_O$ orderings were identified along the [100] and [110] directions in the superlattice system. Our findings have significant implications in strongly correlative solid-state ionics, particularly where fast low-$T$ ionic transport is required, such as in all-solid-state batteries and fuel cells.

## Methods

### Growth of epitaxial BCFO thin films and crystal structure characterization

Pellets with 10% bismuth excess were synthesized by mixing $Bi_2O_3$ (99.9%), CaO (99.995%) and $Fe_2O_3$ (99.9%) powders (Sigma-Aldrich) with different $x_{Ca}$s. The pellets were pressurized to form 1-inch-diameter button-shaped targets. They were calcinated at 800 °C for 8 h in ambient conditions. After the calcination, the pellets were ground into fine powders, and formed into the same shape as the previous targets. Then, they were sintered at 850 °C for 8.5 h under ambient conditions. The epitaxial BCFO thin films ($x_{Ca} = 0.1-0.6$) were

deposited on a $SrTiO_3$ (001) substrate (CrysTec GmbH) using pulsed laser deposition with a KrF excimer laser ($\lambda = 248$ nm). The heater $T$ during film growth was 665 °C in an oxygen environment of 0.07 Torr. Laser fluence and repetition rate were set to be -1 J cm$^{-2}$ and 10 Hz. All the films were in-situ cooled down to room $T$ at a rate of 10 °C min$^{-1}$ under an oxygen environment of 500 Torr. The $c$-axis lattice parameters of the as-grown BCFO thin films were characterized using a four-circle X-ray diffractometer (PANalytical X'Pert PRO MRD) with Cu $K_{\alpha 1}$ radiation. We measured $2\theta-\omega$ X-ray scans from 10° to 60° at an interval of 0.1°. We also performed reciprocal space maps and line scans using a synchrotron source (Beamline 3A, PLS II) in Pohang Accelerator Laboratory.

### Fabrication of BCFO micro-channel devices

A device pattern of eight channels (400 μm × 50 μm) was printed on each as-grown BCFO thin film for all $x_{Ca}$s by UV lithography. AZ5214E photoresist (AZ Electronic Materials) was used as a positive printing material to remove UV illuminated areas. The patterned samples were dry-etched by Ar$^+$-ion milling using a DC ion beam source of 2.5-cm diameter. The unexposed areas of the photoresist were slowly etched at a rate of - 1 nm min$^{-1}$ by application of an argon beam at an acceleration voltage of 750 V and a beam current of 5 mA to prevent surface damage. During the etching process, samples were preserved on the stage by continuously circulating cooling water to obstruct thermal decomposition and degradation. We deposited a $LaAlO_3$ capping layer of 10-nm thickness on the entire surface of the samples using pulsed laser deposition after finishing the etching process, so that all areas of the patterned samples including channel edges were protected from reaction with external oxygen ions in air. The capping layer was grown at a heater $T$ of 650 °C in an oxygen pressure of 0.01 Torr with a laser

fluence of ~ 1 J cm$^{-1}$ and a repetition rate of 2 Hz. After the epitaxial growth, all samples were cooled down at a rate of 10 °C min$^{-1}$ in an oxygen environment of 500 Torr. We patterned eight pairs of electrodes at both ends of the BCFO channels using UV lithography. We removed the LaAlO$_3$ capping layer on the electrode areas using Ar$^+$-ion milling to directly contact platinum electrodes to the BCFO surface. After that, we in-situ deposited platinum by DC magnetron sputtering operated at a power of 25 W under an argon pressure of 5 mTorr.

## Optical visualization and electrical measurement

The patterned BCFO thin films were attached to a custom-made heater, which was mounted on an optical microscope stage (HNM005, HiMax Tech). The custom-made heater was constructed by putting Ni−Cr alloy wire and molding alumina cement between alumina plates. A K-type thermocouple was built on this heater surface to measure the $T$ of samples. A power supply of 30 W was applied to this heater to control $T$ up to 400 °C. We used a 10x long-working-distance objective lens and a 10x eyepiece lens to magnify the patterned BCFO thin films and to secure 34-mm spacing between BCFO films and lens. A color CCD (CVC-5520, Veltek International, Inc.) camera with a pixel resolution of 720 × 480 was equipped on the optical microscope to record the electrical forming processes of the BCFO thin films at a rate of 6 frames per second. Two gold-coated probe tips on positioners (MS Tech) were connected to the platinum electrodes of the BCFO channel. A voltage source (Keithley 230, Tektronix) and a current meter (Keithley 2000, Tektronix) were employed to perform the electroforming process on the BCFO thin films. An external voltage of 25 V was applied at high $T$s. The electrical channel current was monitored simultaneously while filming the video.

## Evaluation of oxygen diffusion mobility via video imaging analysis

The still images of the electroforming process were extracted from the video material using Adobe Premier Pro CC 2015 software. Each still image contained R (red), G (green) and B (blue) information. Our analysis was focused on the center horizontal line of each BCFO channel, which had a width of ~5.6 μm with five vertical pixels. All of the lines were stacked in elapsed time order. We performed the same analysis process every $T$. The electroforming process was performed on different channels of each BCFO thin film for each $T$, and each component of the RGB color information was interpolated to a reference value, which was determined from the average RGB component of the initial sample color and electrically-formed sample color. The reference value was dependent on $x_{Ca}$, because the optical darkness of the electrically-formed state could be changed based on hole carrier concentrations. The spatial boundary between the dark and dark-yellow regions represents the motion trajectory of collective $V_O$ migration. The trajectory, $z(t)$ was constructed by finding the minimum of differentiation with respect to $t$ at each $z$. Each trajectory was fit into our model to obtain the $t_0$ and $\mu$. Since the dark region was more highly conducting than the brighter regions, the applied voltage ($V$) was mainly across the brighter insulating region with a length of $L - z(t)$. This allows us to construct the equation of motion for $z(t)$, using the fact that the instantaneous velocity of the trajectory is proportional to the electric field $E(z, t)$ across the bright region, giving $\frac{dz(t)}{dt} = \mu E(z, t) = \frac{\mu V}{L - z(t)}$. The exact solution of the partial differential equation is the fitting function in the main text with undetermined parameters $t_0$ and $\mu$.

## Impedance spectroscopy of N$_2$ annealed BCFO films

The ionic conductivity was evaluated using AC impedance spectroscopy (Solartron 1260A Impedance Analyzer and 1296A Dielectric Interface System, AMETEK). Before the measurement of AC impedance, all BCFO films were annealed at 635 °C in a N$_2$ gas environment of 10 mTorr for 24 h, to suppress electrical conduction[22] (see

Supplementary Note 3). The entire area of the BCFO film was covered by a LaAlO$_3$ capping layer (10 nm in thickness) to form the same geometry as was used for optical visualization. Interdigitated platinum electrodes with the well-known ion-blocking property were deposited on the BCFO film surface after etching the LaAlO$_3$ on the electrode regions. AC voltages of 100 mV at frequencies ranging from 1 MHz to 1 Hz were applied across the interdigitated electrodes. All measurements were performed in a vacuum environment (MHCS622-V/G, MicroOptik). An equivalent circuit model (Supplementary Fig. 3b) was applied to interpret the experimental data using the commercial software, ZVIEW.

## Transmission electron microscope (TEM)

Electron transparent cross-sectioned samples for transmission electron microscopy were prepared with an FEI Helios NanoLab 600i focused ion beam (FIB) instrument operated at accelerating voltages of 30 and 5 kV. Scanning transmission electron microscopy (STEM) imaging was performed on a probe-corrected FEI Titan Themis microscope operated at 300 kV. The experiments were carried out with a probe convergence semi-angle of 18 mrad and collecting semiangles of 10−20 mrad and 70−190 mrad for the annular-bright-field (ABF) and high-angular dark-field (HAADF) STEM detectors, respectively. Atomic resolution imaging of the cation sublattices in the perovskite BFCO films is best accomplished via HAADF-STEM imaging, as its signal is proportional to $Z^n$ ($n \approx 1.6$–2.0). ABF-STEM imaging, whose signal is proportional to $Z^m$ ($m \approx 1/3$), allows visualizing both the anion and cation sublattices simultaneously. To correct for the scan distortions, time series consisting of 10 frames (2048 × 2048 pixels) were acquired and averaged by rigid and non-rigid registration using the Smart Align software[36]. The processing of the ABF- and HAADF-STEM images was performed in MATLAB, using custom-developed scripts. To remove the low-frequency background intensity variations, mostly caused by specimen surface contamination, we performed a morphological opening using a structuring element[37]. The best results were obtained using a disk structuring element with a sufficiently large radius (e.g., a few interatomic distances) and by smoothing the obtained result with a 2D Gaussian filter (with the same radius as the structuring element). In this way, we were able to extract the background of the image which contained the low spatial frequency (~15−20 Å$^{-1}$) information due to specimen surface contamination. The background was then subtracted from the raw data to obtain the background-corrected signal. The background-corrected signal was finally denoised using a custom-developed nonlinear filtering algorithm based on the method proposed by Du[38]. The obtained results provided the denoised and background-corrected datasets. Due to the finite lateral size of the electron probe, deconvolution processing of the ABF-STEM images was used to eliminate the effects of the probe function from the high-resolution STEM signals[39]. In this work, the deconvolution process was performed using a conventional blind deconvolution algorithm (MATLAB's "deconvblind" function), using an initial estimate of the point-spread function as calculated from the microscope's parameters (i.e., acceleration voltage and beam convergence semi-angle). The contrast of all ABF images presented here was inverted for visual clarity so that the peaks corresponding to atomic columns appear as local intensity maxima in the images. The atomic column positions in the ABF- and HAADF-STEM images were fitted by means of a center of mass peak-finding algorithm and subsequently refined by solving a least-squares minimization problem using the Levenberg−Marquardt algorithm. This iterative refinement was carried out using a custom-developed MATLAB script that makes use of 7-parameter two-dimensional Gaussians. The fitting allows quantitative estimation of the atomic column peak intensities and their positions with picometer precision[19,40]. In addition, the positions of the oxygen-vacancy channels displaying zero intensity were manually added. Lastly, a quantitative analysis of the lattice parameters and the

Fe–Fe distances at the oxygen-deficient layers was performed using a custom-developed peak-pair analysis[41] MATLAB script. Electron diffraction (ED) was performed using a JEOL 2200FS TEM/STEM microscope operated at 200 kV.

## Computational methods

All DFT calculations were performed using the projector-augmented wave (PAW) method[42], as implemented in the Vienna Ab-initio Simulation Package (VASP) (refs. [43], [44]). PAW pseudopotentials in the standard VASP database were applied to Bi, Ca, Fe, and O atoms. A plane-wave basis set with a kinetic energy cutoff of 600 eV was used for wavefunction expansion. To simulate superlattice models, various supercells were constructed with an extension of a basic $2 \times 2 \times 2$ ($\sqrt{2} \times \sqrt{2} \times 1$) 40-atom supercell for cubic perovskite including 5 atoms (Pnma distorted perovskite including 20 atoms). For all calculated cells, the Brillouin zone integration was applied, corresponding to a $2 \times 2 \times 2$ grid with $\Gamma$ of $k$-points meshes in the basic supercell containing 40 atoms. Here we applied a PBEsol (Perdew–Burke–Ernzerhof revised for solids[45]) + $U$ (ref. [46]) of about $U$–$J$ = 5.5, 7.5, and 5.5 eV to improve the description of the Ca, Bi, and Fe 3$d$ states in the BiFeO₃ and BCFO. All atomic positions were fully relaxed until the Hellmann–Feynman force on each atom was within 0.01 eV Å$^{-1}$.

## Crystal symmetry modeling of the BCFO films grown on SrTiO₃

Multiferroic (un)doped BiFeO₃ thin films are well-known to present many lattice instabilities resulting in several low-energy phases, such as $Pc$, $Cm$, $Pna21$, $Cc$, $Pnma$, and $R3c$. The stability of the low-energy phases can be determined by the degree and type of the structural distortions, such as antiferrodistortive patterns with in-phase ($M_3^+$) and antiphase ($R_4^+$) rotation modes of the FeO₆ octahedra, and ferroelectric ($\Gamma_4^-$) and antiferroelectric patterns ($X_5^-$, $X_5^+$, $M_5^-$, $M_5^+$, and $R_5^+$). In our experiments, the as-grown BCFO thin films were measured to have a pseudo-tetragonal paraelectric structure, but their space groups could not be unambiguously assigned. Since the BCFO thin films were not ferroelectric, the $Pnma$ space group with no ferroelectric $\Gamma_4^-$ pattern was selected as the space group to model the BCFO crystal structure (ref. [47]). In addition, to maintain consistency with the experiment, it was assumed that the antiferrodistortive in-phase rotation direction was stacked along the direction normal to the SrTiO₃ substrate. The spin–spin exchange interaction between the $Fe^{3+}$ ions was set to be G-type antiferromagnetic, which is the most stable. To simulate the thin film structure, the in-plane lattice constant of the BiFeO₃ $Pnma$ phase was fixed to the experimental SrTiO₃ lattice constant (3.905 Å). The $c/a$ ratio of pure BiFeO₃ was calculated to be 1.01, less than the experimental 1.04. Yet, $c/a$ ratios less than 1 were obtained for various high $x_{Ca}$ BCFO films, in good agreement with the experimental data. Moreover, the band gap of the calculated $Pnma$ BiFeO₃ phase (about 2.26 eV) was consistent with the experimental optical band gap of about 2.74 eV (ref. [48]).

## Young's modulus using atomic force microscope

To evaluate the elastic modulus of the BCFO films, force-distance curves were obtained using an atomic force microscope (MFD-3D infinity, Asylum Research, Oxford Instruments) at room $T$ under an ambient atmosphere. A platinum coated AFM tip (HQ:NSC35/Pt, MikroMasch) was used to press the sample surface. When the AFM tip closely approached the sample surface, van-der Waals forces attract the AFM tip to the sample surface. As the piezo-stage goes upward in the approach mode, the AFM tip is deflected by the force applied to the sample surface. To calibrate the applied force on the sample surface, we use a sapphire substrate (CrysTec GmbH) as a non-deformable reference sample (~470 GPa) to obtain optical lever sensitivity. The sensitivity is defined as $\Delta Z/\Delta V_d$, where $Z$ is the position of the piezo-stage, and $V_d$ is the deflection voltage in the photodetector. The spring constant of the AFM tip, $k$, was evaluated by the thermal noise spectrum method. The applied force was calculated using Hooke's law, $F = -kd$, where the $d$ is the deflection distance. The Young's modulus was extracted using the Derjaguin–Muller–Toporov (DMT) model. The model was proposed for the force-distance measurements that involves relatively weak adhesion forces and small tip radius curvature (ref. [49]). The DMT model is expressed as $F - F_{adh} = 3/4E^* \sqrt{Rd^3}$ where $E^*$, $R$ and $d$ are reduced Young's modulus, the radius of the curvature of the tip apex, and the deformation depth, respectively. The reduced modulus is related to the Young's modulus via the following relation: $1/E^* = (1 - v^2)/E + (1 - v_{tip}^2)/E_{tip}$ where $v$ and $E$ are the Poisson's ratio and the effective Young's modulus of a sample, respectively (e.g. $v$ ~ 0.25 for BiFeO₃). We used the values of the AFM tip (i.e., platinum coated silicon tip: $E_{tip}$ ~ 170 GPa and $v_{tip}$ ~ 0.27). The deformation depth of the BCFO film was measured using the difference in deflection of the reference sample and the BCFO film. The $R$ was commercially known to be 30 nm. We performed force-distance curve measurements at multiple points (~30 pts) on each BCFO film surface.

## Reporting summary

Further information on experimental design is available in the Nature Research Reporting Summary linked to this paper.

## Data availability

The data that support the findings of this study are available from the corresponding authors upon reasonable request.

## Code availability

All total energy and electronic structure analysis data contained in this work were generated with the VASP code, which is available for a license fee from https://www.vasp.at. Custom MATLAB codes are available upon request to the corresponding authors.

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

## Acknowledgements

We thank Prof. Yong Woon Kim and Prof. Kyung-Jin Lee of KAIST and Prof. Sidney R. Nagel of the University of Chicago for discussing the mechanism. C.-H.Y. acknowledges the support of a National Research Foundation (NRF) of Korea grant funded by the Korean Government through the Creative Research Center for Lattice Defectronics (grant no. NRF-2017R1A3B1023686). C.-H.Y. and Y.-H.K. acknowledge the support of Center for Quantum Coherence in Condensed Matter (2016R1A5A1008184). Y.-H.K. acknowledges the support of a NRF of Korea program (2019M3D1A1078302). M.C. and M.D.R. acknowledge support by the Swiss National Science Foundation under Project No. 200021-175926.

## Author contributions

J.S.L., H.-S.P., and J.S. prepared the samples and devices with basic characterizations and performed real-time electroforming processes with color tracing analyses. M.C. and M.D.R. carried out TEM measurements. H.-H.N., J.L., J.J., and Y.-H.K. performed first-principles DFT calculations. Y.-J.K., H.-S.P., J.S.L., Y.Y., and T.Y.K. performed x-ray diffraction and analysis for structural characterization. H.-S.P. and J.S. perform oxygen isotope diffusion experiments. J.S.L., Y.-H.K., M.D.R., and C.-H.Y. led the manuscript preparation with contributions from all authors.

## Competing interests

J.S.L. and C.-H.Y. have patents (10-2182181-0000, registered in Korea; US 11,211628 B2, registered in the USA) and patent application (EU19197916.0), which disclose bismuth calcium ferrites for electrolyte having high oxygen ionic mobility. The remaining authors declare no competing interests.
