## [Peer Review File · Nature Communications]

Critical ionic transport across an oxygen-vacancy ordering transitionREVIEWER COMMENTS

Reviewer #4 (Remarks to the Author):

The authors provided many good discussions in the revised manuscript. The background on oxygen ordering and transport in other perovskites definitely clarified the uniqueness of the vacancies in BCFO. The clarification of oxygen interaction distance warranted the usage of the small unit cell.

However, I found the arguments and terminologies used in this paper often confusing.

1. Clarify the phase and phase boundary.

1.1) There is no clarification of the two phases in the data. The author claimed they are “isostructure phases” in the abstract. Please clarify what are “isostructure phases”? How did they separate one phase from another? If they are simply one phase for being “isostructures”, should these be called grain boundaries instead of phase boundaries?

1.2) Why do the oxygen-deficient planes show up as phase boundaries?

2. Clarify “competition”.

2.1) The second sentence in the abstract said “However, the lack of a lattice system that exhibits competing defect orderings has hindered the study of the feasibility.” First of all, this sentence is extremely confusing. These are crystalline structures, why there is “the lack of a lattice system”? Oxygen vacancy ordering occurs because the ordered structures have lower energy. What are the factors that are competing here?

2.2) In the main text, the authors focused on discussing the competition as the competition of VO channel orders between $\langle 100 \rangle$ and $\langle 110 \rangle$. However, the nature of this competition is not clear. The authors used DFT to show if these structures exist, the oxygen migration barrier is lower but did not show the energy competition of these structures.

3. The authors added a new paragraph to discuss elastic softening. It's totally out of place from the rest of the discussion related to oxygen vacancy, ordering, and diffusion. At the end of this discussion, the author suggested that phase boundary exists in the composition of $x_{Ca}=0.45$. However, on page 7, the author said the oxygen-deficient layers appear in all three compositions (“appear periodically every ~ 6 ($x_{Ca} = 0.3$), 4 ($x_{Ca} = 0.45$), as well as 3 ($x_{Ca} = 0.6$) perovskite blocks along the $[001]$ growing direction”). These statements seem to conflict with each other.

I think both the science and the writing of this paper need to be improved for the broader readership of Nature Communication.

Response Letter

Reviewer #4-General

Reviewer #4 (Remarks to the Author):

The authors provided many good discussions in the revised manuscript. The background on oxygen ordering and transport in other perovskites definitely clarified the uniqueness of the vacancies in BCFO. The clarification of oxygen interaction distance warranted the usage of the small unit cell.

However, I found the arguments and terminologies used in this paper often confusing.

Response #4-General :

We sincerely appreciate the referee's comments, which significantly improve the manuscript by clarifying the meanings of confusing terminologies and ambiguous expressions. The abstract and the second paragraph of the introduction are considerably revised.

Reviewer #4-1-1

1. Clarify the phase and phase boundary.

1.1) There is no clarification of the two phases in the data. The author claimed they are "isostructure phases" in the abstract. Please clarify what are "isostructure phases"? How did they separate one phase from another? If they are simply one phase for being "isostructures", should these be called grain boundaries instead of phase boundaries?

Response :

It is an important question related to our main argument. We are sorry that we couldn't deliver the meaning of the terminologies more clearly. We will explain what we intended in the following lines.

The compositional phase boundary is not a grain boundary in real space but it is used to refer to a phase transition between the two oxygen-vacancy ordering (along $\langle 100 \rangle$ or $\langle 110 \rangle$) phases as a result of varying the Bi/Ca ratio in $\text{Bi}_{1-x}\text{Ca}_x\text{FeO}_{3-\delta}$. More concretely, the compositional phase boundary refers to the Ca doping ratio of $x_{\text{Ca}}=0.45$. The $\langle 100 \rangle$ -type oxygen-vacancy ordering appears in compounds below $x_{\text{Ca}} < 0.45$ while the $\langle 110 \rangle$ -type ordering emerges in Ca-rich compounds ($x_{\text{Ca}} > 0.45$).

One important example of the similar use of the terminology is the ferroelectric morphotropic (or compositional) phase boundary (MPB) that separates the rhombohedral phase and the tetragonal phase by control of the Zr/Ti ratio (e.g, in $\text{PbZr}_{1-x}\text{Ti}_x\text{O}_3$ ceramics). It is a phase boundary in a phase diagram defined in two parameters space (composition and temperature) (see Fig. R1a).

The observation of an oxygen-vacancy ordering transition between $\langle 100 \rangle$ versus $\langle 110 \rangle$ channels and the suppression of migration activation energy have an analogy to the ferroelectric MPB and a large enhancement of dielectric constant and piezoelectric coefficient (Fig. R1b).

Although our original motivation for the use of "compositional phase boundary" is explained, the use may cause confusion. **So, we replace the terminology by "compositional phase transition point" throughout the paper. Hopefully, the changes improve the manuscript in terms of clarity for a broad readership.**

Moreover, we rewrite several sentences in the abstract or the introduction to deliver the meaning more clearly as below:

“Our demonstration relies on the finding that a compositional phase transition occurs by varying Ca doping ratio across $x_{Ca} \approx 0.45$ between two structural phases with oxygen-vacancy channel ordering along $\langle 100 \rangle$ or $\langle 110 \rangle$ crystal axis, respectively.” in the abstract;

“Here, we propose a way to significantly lower E_A by controlling chemical doping to find the compositional phase transition point that separates two different structural phases of oxygen vacancy (V_O) order.” in the introduction.

Figure R1: A ferroelectric morphotropic phase boundary. a, Phase diagram with a competition of rhombohedral [111] phase and tetragonal [001] phase. b, Enhanced dielectric constant and electromechanical coupling at the compositional phase boundary. † Cross, L. E. *Ferroelectric Ceramics-Tutorial Reviews, Theory, Processing and Applications*; Setter, N. & Colla, E. L. ed. 1 (Birkhauser Verlag, Basel, 1993). ‡ Jaffe, B., Cook Jr., W. R. & Jaffe, H. *Piezoelectric Ceramics* (Academic Press, London, 1971).

Although these two phases have different oxygen vacancy orderings within the (001) oxygen-deficient planes, it is challenging to distinguish them by using conventional X-ray diffraction because they have similar structures except for vacant oxygen-ion sites. Careful high-sensitivity measurements are required as we did in the paper. The two structures expressed by isostructures are very similar but it is not identical to each other.

We understand the point the referee mentioned. The boundaries between isostructures are reminiscent of grain boundaries between two different orientation states. We removed the use of "isostructures" from the revised manuscript in pursuit of a more scientifically valid expression. We appreciate the referee's completeness in using scientific terminologies.

Reviewer #4-1-2

1.2) Why do the oxygen-deficient planes show up as phase boundaries?

Response :

As addressed in the response to the comment above, the compositional phase boundary is neither a grain boundary nor an interface in real space. The oxygen-deficient planes are only parts of the crystal structures.

Since the oxygen-deficient planes are important routes for ionic transport, we would like to explain more details of the planes.

Interestingly, our BCFO films exhibit superlattice structures in a self-assembled way (Fig. R2). The oxygen-deficient (001) planes appear periodically along c axis. The higher the Ca doping ratio (x_{Ca}), the shorter the spacing between adjacent planes (d in unit cells) according to the empirical rule of $d \approx 1.5/x_{Ca}$ (Fig. R3). It indicates the oxygen vacancy density within a single deficient layer is constant irrespective of different Ca doping ratio.

In addition to controlling the inter-spacing between adjacent oxygen-deficient planes, the Ca-doping ratio (x) changes the c/a ratio (the tetragonality factor of unit cell) of oxygen-deficient planes. Because the effective ionic radius of Ca^{2+} ion is smaller than that of Bi^{3+} ion, the c/a ratio becomes smaller with increase of x_{Ca} in the situation that the in-plane pseudocubic lattice constant (a) is clamped to that of the substrate. This structural change of c/a induces a phase transition in oxygen-vacancy configuration within the oxygen-deficient planes.

Figure R2: STEM image for a cross-sectional plane of $Bi_{0.8}Ca_{0.2}FeO_{3-d}$. **a**, Color-coded HAADF micrograph in the [010] zone axis. The growth direction is from bottom to top. The horizontal white lines indicate the position of the dark-layers at each side of the image, appearing every 7 or 8 perovskite blocks. **b**, Corresponding FFT indexed within the pseudocubic reference system. **c**, Line profiles along the [001] and [100] directions of the FFT. The satellite spots (SL) due to the superstructure are visible

along the [001] direction at a spacing of 0.35 nm^{-1} , while an additional spot at 1.28 nm^{-1} (marked by the black arrow) corresponding to a doubling of the perovskite periodicity is observed in the profile taken along the [100] direction. Adapted from [Ref. M. Campanini, et al. Nano Letters 18, 717 (2018)].

Figure R3: Structures of as-grown BCFO films on SrTiO_3 substrates. **a**, X-ray 2θ - ω scans at room T for samples with varying x_{Ca} (0.1 ~ 0.6). Dashed line represents the (001) peak of SrTiO_3 . BCFO films at $x_{\text{Ca}} = 0.2$ or above have superlattice peaks (green lines, SL \pm), indicating the periodic appearance of V_O layers along the c axis. **b**, c -axis lattice parameter versus x_{Ca} . The red line denotes the linear interpolation between the lattice parameters of BiFeO_3 and $\text{CaFeO}_{2.5}$ according to Vegard's law. Error bars were estimated to be 1/5 of the FWHMs of the BCFO film peaks. **c**, Periodicity of ordered V_O layers versus x_{Ca} . Error bars were estimated from the FWHMs of the SL \pm peaks. Red line denotes an empirical rule, i.e., $n_{\text{period}} = 1.5/x_{\text{Ca}}$. As the distance between V_O layers becomes closer, it tends to deviate from the empirical rule. BCFO films of $x_{\text{Ca}} = 0.2$ (0.3) have $n_{\text{period}} = 7$ or 8 (5 or 6) unit cells, but for $x_{\text{Ca}} = 0.6$, most of the V_O layers are ordered with a periodicity of 3 unit cells. Adapted from Supplementary Fig. 1 in the manuscript.

Reviewer #4-2-1

2. Clarify “competition”.

2.1) The second sentence in the abstract said “However, the lack of a lattice system that exhibits competing defect orderings has hindered the study of the feasibility.” First of all, this sentence is extremely confusing. These are crystalline structures, why there is “the lack of a lattice system”? Oxygen vacancy ordering occurs because the ordered structures have lower energy. What are the factors that are competing here?

Response :

We find that the oxygen ionic configuration within the oxygen deficient layer undergoes a phase transition as a result of varying c/a ratio that is implicitly controlled by Ca doping ratio. The phase transition of oxygen-vacancy ordering occurs at a compositional phase transition point, i.e., $x_{\text{Ca}}=0.45$. We thought that this kind of lattice system having such a phase transition point induced by chemical substitution is uncommon.

We understand the point the referee raised. Since our sample is a crystal, various grain boundaries or twin walls between simply different orientation states are possible. We think the question stems from a misunderstanding of the originally used compositional phase boundary. The phrase is removed during shortening the abstract according to the journal-style length limitation. It also seems reasonable to

remove the sentence as there are a few examples of phase transitions driven by chemical doping, such as in aliovalent doped ferrites.

Reviewer #4-2-2

2.2) In the main text, the authors focused on discussing the competition as the competition of VO channel orders between $\langle 100 \rangle$ and $\langle 110 \rangle$. However, the nature of this competition is not clear. The authors used DFT to show if these structures exist, the oxygen migration barrier is lower but did not show the energy competition of these structures.

Response :

We are deeply grateful for the valuable comments that the referee pointed out for a deeper understanding of the V_O -channel orientation depending on Ca doping or the c/a ratio. We explain the competition in the following lines for an intuitive understanding of the channel selection depending on Ca doping and consequently the c/a ratio.

It was experimentally measured that the c/a ratio decreases as x_{Ca} increases. It can be understood based on the concept of tolerance factor, $t = (r_A + r_O) / \sqrt{2}(r_B + r_O)$, which describes the degree of atomic distortion when A-O and B-O bondlengths change. The c -axis lattice constant shrinks more at high Ca concentration ($x_{Ca} = 0.75$) than at low Ca concentration ($x_{Ca} = 0.25$). From a polygonal structure point of view, as shown in Fig. R4a, the $[100]$ and $[110]$ channels are characterized by two tetrahedral relaxations, either symmetric or antisymmetric relaxation patterns.

A smaller c -axis lattice constant with increasing x_{Ca} causes a larger apical oxygen shift in the tetrahedron, as depicted in Fig. R4b. In the symmetric $\langle 100 \rangle$ configuration, the two apical oxygens of adjacent tetrahedra are closer to each other than in the $\langle 110 \rangle$ case, resulting in a larger energy penalty.

Figure R4: Schematic diagrams of symmetric and antisymmetric patterns of tetrahedral relaxations for two values of x_{Ca} : $[100]$ channel order with large c/a ($x_{Ca} = 0.25$) and $[110]$ channel order with relatively small c/a ($x_{Ca} = 0.75$). **a, (001) plane view of oxygen-deficient planes. **b**, (100) plane view of oxygen-deficient planes. The red balls represent oxygen vacancies forming the tetrahedra of oxygen anions (expressed by green balls). The smaller c -axis lattice constant**

causes a larger tetrahedral relaxation, which selects the antisymmetric [110] pattern to avoid bumping of two adjacent tetrahedra. Adapted from Supplementary Fig. 9.

As asked by the referee, we included DFT calculation results for more quantitative energy comparison between the [110] and [100] channel ordering configurations. We compared the two energies at two selected x_{Ca} for three in-plane lattice constants ($2a$). It can be seen that the higher Ca doping ratio is more likely to favor the [110] channel order. All the BCFO films are grown in SrTiO₃ substrates, so the in-plane lattice parameters are fixed to $2 \times 3.905 = 7.81 \text{ \AA}$. At this value of in-plane lattice parameter, the BCFO at $x_{Ca}=0.25$ prefers to stabilize the [100]-channel order while the BCFO at $x_{Ca}=0.75$ prefers the [110]. The theoretical evaluation of energy competition shows an excellent agreement with the experimental results.

Figure R5: Energy comparison of the two competing structures. The energy value represents the relative energetics, $E([100]) - E([110])$ at low and high Ca concentrations ($x_{Ca} = 0.25$ and 0.75). $E([100])$ and $E([110])$ are total energies of [100]- and [110]-aligned superlattices, respectively.

We would like to thank the referee for the highly-professional comment and for the deep physical insight of the results obtained in the work. We include the energy comparison results in Supplementary Fig. 14 of the revised manuscript. We also add the following paragraph in the associated text of the Supplementary Information : “Fig. S14 shows DFT calculation results for energy comparison between the [110] and [100] channel ordering configurations. We compared the two energies at two selected values of $x_{Ca} = 0.25$ and 0.75 for three in-plane lattice constants ($2a$). It can be seen that the higher Ca doping ratio is more likely to favor the [110] channel order. All the BCFO films are grown in SrTiO₃ substrates, so the in-plane lattice parameters are fixed to $2 \times 3.905 = 7.81 \text{ \AA}$. At this value of in-plane lattice parameter, the BCFO at $x_{Ca}=0.25$ prefers to stabilize the [100]-channel order while the BCFO at $x_{Ca}=0.75$ prefers the [110]. The theoretical evaluation of energy competition shows an excellent agreement with the experimental results.”. To link the calculation result to the main text, we add the sentence “More quantitative energy comparisons between the two competing V_O -channel phases were studied at selected x_{Ca} and in-plane lattice parameters (Supplementary Fig. 14).” in the revised manuscript.

Reviewer #4-3

3. The authors added a new paragraph to discuss elastic softening. It's totally out of place from the rest of the discussion related to oxygen vacancy, ordering, and diffusion. At the end of this discussion, the author suggested that phase boundary exists in the composition of $x_{Ca}=0.45$. However, on page 7, the author said the oxygen-deficient layers appear in all three compositions ("appear periodically every ~6 ($x_{Ca} = 0.3$), 4 ($x_{Ca} = 0.45$), as well as 3 ($x_{Ca} = 0.6$) perovskite blocks along the [001] growing direction"). These statements seem to conflict with each other.

Response :

The tetragonality factor c/a ratio is continuously controlled by changing Ca doping ratio across the phase transition point. The Ca-doping-ratio dependence of elastic constant clearly shows the existence of a phase transition point, through elastic softening at the compositional phase transition point ($x_{Ca} = 0.45$). Following the referee's comment, we minimize the technical description of obtaining the elastic constant to improve readability in the main text. We agree that the technical description diverts our attention somewhat from the main focus. Since the details are already addressed in the Methods, the shortening does not cause significant loss of meaning.

Once again, the phase boundary doesn't mean the oxygen deficient planes. The oxygen deficient planes as a part of real-space crystal structure are found in all the BCFO compounds regardless of different x_{Ca} . So, the statements don't conflict with each other.

Reviewer #4-4

I think both the science and the writing of this paper need to be improved for the broader readership of Nature Communication.

Response :

We greatly appreciate the referee's constructive comments on our manuscript. We made an effort to improve the manuscript by clarifying the meaning of confusing terminologies and ambiguous expressions for the broader readership.

REVIEWERS' COMMENTS

Reviewer #4 (Remarks to the Author):

The authors have addressed my concerns. The logic of the paper is much more clear now.

REVIEWERS' COMMENTS

Reviewer #4 (Remarks to the Author):

The authors have addressed my concerns. The logic of the paper is much more clear now.

Response:

We appreciate the reviewer's opinion.